# Exploring the Developmental Potential of Human Germinal Vesicle Oocytes Aiming at Fertility Preservation: Can We Increase the Yields of Competent Oocytes through IVM Combined with Vitrification?

**DOI:** 10.3390/jcm11061703

**Published:** 2022-03-19

**Authors:** Xia Hao, Jessie Phoon, Lina Barbunopulos, Mona Sheikhi, Arturo Reyes Palomares, Kenny A. Rodriguez-Wallberg

**Affiliations:** 1Department of Oncology and Pathology, Karolinska Institutet, Solna, SE-171 76 Stockholm, Sweden; xia.hao@ki.se; 2Laboratory of Translational Fertility Preservation, BioClinicum, SE-171 64 Stockholm, Sweden; 3Department of Reproductive Medicine, Division of Gynecology and Reproduction, Karolinska University Hospital, Novumhuset Plan 4, SE-141 86 Stockholm, Sweden; jess_phoon@yahoo.com (J.P.); lina.barbunopulos@vgregion.se (L.B.); mona.m.sheikhi@gmail.com (M.S.)

**Keywords:** fertility preservation, immature oocyte, germinal vesicle, vitrification, in vitro maturation, fertilization, blastocyst

## Abstract

The rescue in vitro maturation (rIVM) of germinal vesicle oocytes (GVs) has been proposed to improve the total number of mature oocytes in women undergoing fertility preservation. Currently, there is no consensus about the clinical utility of this practice, and heterogeneity in the protocols used may influence the final outcomes. This study investigated the developmental potential of mature metaphase II (MII) human oocytes obtained from GVs after rIVM and the impact of applying vitrification at different timepoints either before or after rIVM. After randomization, oocytes were assigned to undergo rIVM and thereafter vitrification or intracytoplasmic sperm injection (ICSI), or to undergo direct vitrification-warming and thereafter rIVM and ICSI. The likelihood of obtaining MII oocytes was just slightly higher in the fresh rIVM group compared to the vitrification-warming-rIVM group. When comparing fresh rIVM that underwent subsequently ICSI, the fertilization and developmental rates up to the blastocyst stage were seen to be reduced in both groups that underwent vitrification either before or after rIVM. Although some blastocysts were obtained in the fresh rIVM-ICSI group, the efficacy of these methods was low overall, suggesting that the further development of protocols for IVM conducted early after denudation is needed to improve the final results of rIVM aiming at fertility preservation.

## 1. Introduction

Increasing numbers of women worldwide are undergoing hormonal stimulation aiming at preserving mature oocytes for future fertility possibilities. The reasons for this are various and include the possibility of future infertility due to gonadotoxic chemotherapy indicated for cancer treatment and the awareness of infertility associated with increasing age [1,2]. The current gold standard technique for fertility preservation in women is the vitrification of mature oocytes, which was endorsed by European and American scientific societies as a clinically established method nearly a decade ago [3,4,5]. A controlled ovarian stimulation (COS) treatment using gonadotropins is required for the development of multiple follicles in one treatment cycle, which is the standard treatment needed to obtain multiple mature oocytes for cycles aimed at in vitro fertilization (IVF). At many centers today, only mature oocytes are vitrified, but after oocyte retrieval around 15% of retrieved oocytes are found immature, and these are discarded without further utilization [6,7].

The in vitro maturation (IVM) of oocytes is an emerging method that may serve as a complementary strategy for fertility preservation. IVM has certain advantages, such as the reduction in the dosage of hormones and reduced costs. If a shorter treatment or no hormonal treatment were conducted, oocyte retrieval could be planned without delaying, which may be an advantage in cases where the prompt initiation of cancer treatment is desired [8,9]. IVM protocols involving minimal stimulation without human chorionic gonadotropin (hCG) have now been established at several medical centers. The efficacy of biphasic IVM with pre-maturation steps has shown promising results [10]. Although they have succeeded, the rates of IVM have not yet reached the outcomes of standard COS for IVF, but they have been significantly improved over time [10]. IVM has been proposed for the fertility preservation of women with cancer who are not recommended hormonal stimulation. It has been also proposed for assisted reproduction in women with a diagnosis of polycystic ovary syndrome (PCOS), where conventional ovarian stimulation is not feasible [3]. However, the developmental competence of IVM oocytes is still low compared to oocytes obtained at mature stage after COS, and further knowledge about its safety is needed [11]. A prospective large cohort study of women undergoing fertility preservation in Sweden reported the cryopreservation of immature oocytes for 22 women with a benign indication and 144 women with cancer [12]. However, most of these oocytes have not been used yet; thus, their developmental potential has not been tested. The use of mature oocytes obtained after IVM also overcomes the risk of re-introducing malignant cells, which could be encountered in the case of ovarian tissue transplantation.

The vitrification of denuded metaphase II (MII) oocytes, after COS cycles, is currently the most successful mode of fertility preservation, and there is a lack of knowledge about cryopreserving oocytes in different immature states. The mature MII oocytes present chromosomes attached to microtubules conforming to the meiotic spindle, a structure vulnerable to temperature changes that could result in an increased risk of chromosomal imbalances [9]. However, germinal vesicle oocytes (GVs) do not present such spindle structures and, during prophase-I, chromatin is protected by the nuclear membrane [9]. In such circumstances, it could be hypothesized that oocyte cryopreservation performed at the GV stage has less impact than at the MII stage [13]. However, after warming, GVs may still need longer support from the surrounding cumulus cells to obtain nutrients and regulatory molecules for maturation [14].

Rescue IVM (rIVM) can be carried out in GVs after oocyte collection and denudation during IVF treatments. These GVs are able to reach the MII stage and be properly fertilized through IVF-intracytoplasmic sperm injection (ICSI), go further in development and reach live birth [15], even when previously frozen and warmed [16]. However, in denuded oocytes, the absence of cumulus cells reduces the chances of reaching an adequate cytoplasmic and nuclear competence and could be a source of epigenetic defects. The concern regarding the application of IVM for fertility preservation is related to the possible heterogeneity between oocytes in the meiotic stage and the cytoplasmic maturity stages, which makes it difficult to choose the proper timing to cryopreserve them. Thus, to date, the clinical use of oocytes from GVs undergoing rIVM is mostly discouraged. However, their use is suitable for research [6].

We conducted a prospective randomized study to investigate the developmental potential of denudated GVs in a step-wise combination with vitrification before or after rIVM. For this study, GVs were donated from female patients 18–39 years of age undergoing IVF-ICSI cycles at the department of Reproductive Medicine of Karolinska University Hospital. Fertilization was attempted through ICSI using donor sperm. Through this strategy, we aimed to assess how vitrification performed at different meiotic states impacts the subsequent maturation, fertilization and embryo developmental potential. After the first step in the selection of an appropriate IVM culture medium from among two candidates, a randomized study was designed. The procedures were all performed by clinical embryologists of our center, who also evaluated the fertilization and subsequent development up until the blastocyst stage, which was one of the main outcomes.

## 2. Materials and Methods

### 2.1. IVM Culture Medium Selection and Fresh vs. Vitrified Stepwise-Randomized Study Design

#### 2.1.1. IVM Culture Medium Selection

According to the power calculation, 98 GVs randomized to each of the two pre-selected media groups would be sufficient to assess a 30% difference in maturation between the two groups. The two pre-selected IVM culture media included SAGE^TM^ In Vitro Maturation Kit (CooperSurgical^®^, Målov, Denmark) (*n* = 49, SAGE group) and MediCult IVM^®^ system (CooperSurgical^®^, Målov, Denmark) (*n* = 49, MediCult group). After 24 h and 48 h of culture, the maturity state of oocytes was evaluated.

#### 2.1.2. Fresh vs. Vitrified Comparison

This comparison was performed in a two-step randomized experiment according to the progress of oocytes’ survival after warming and maturity. The first step of randomization was performed at the oocyte collection and a total of 102 GVs were randomly assigned to undergo fresh rIVM (*n* = 75) or be immediately vitrified, warmed and undergo subsequent IVM (*n* = 27). IVM culture was performed in all GVs by using the SAGE IVM medium for 24 h, as this was the medium that showed a significantly higher maturation outcome in the previous step for the culture medium selection, and there was no evidence at that step of further optimization in IVM results with prolonged culture after 24 h. In the second step, MII oocytes obtained from fresh rIVM were further randomly assigned to the fresh rIVM-vitrification-warming-ICSI (*n* = 16) group or the fresh rIVM-fresh ICSI (*n* = 17) group (control). Mature oocytes in all three groups were fertilized by cryopreserved-thawed spermatozoa from a screened sperm donor through ICSI. Their fertilization status was checked 16–20 h after ICSI; subsequent embryo development was evaluated by time-lapse (Primovision™, Vitrolife, Midtjylland, Denmark) over a six-day period.

### 2.2. GVs and Spermatozoa Resources

Immature GVs were donated from female patients who underwent IVF-ICSI cycles at the department of Reproductive Medicine, division of Gynecology and Reproduction, Karolinska University Hospital. Patients aged between 18 and 39 years with no known ovarian pathology were anonymously included in the study, as required by our ethical application. Thawed spermatozoa used for ICSI were from a single donor with proven fertility and remained cryopreserved at the center. The samples were also anonymized. The ethical approval for this study was granted by the Regional Ethics Committee of Stockholm (Dnr 2010/549-31/2 Amendment 2012/66-32).

### 2.3. Oocyte Collection and Denudation

Oocyte collection was performed using a single-lumen aspiration needle under transvaginal ultrasound guidance. All developed follicles were aspirated and subsequently follicular fluid was collected in tubes to be processed in the IVF laboratory. Cumulus oocytes complexes (COCs) were retrieved under a stereomicroscope and collected for further processing.

The denudation of COCs was performed by mechanical disaggregation after the brief exposure of hyaluronidase (40 IU/mL) in a phosphate-buffered culture medium. After removing granulosa cells, the assessment of the maturity state of oocytes was performed. GVs were used for the following experiments.

### 2.4. rIVM of GVs

Two different protocols from the same commercial provider (CooperSurgical^®^) for the rIVM of GVs were applied in the IVM culture medium selection, as follows:

SAGE^TM^ In vitro Maturation Kit: denudated immature oocytes were cultured in pre-equilibrated oocyte maturation medium supplemented with 75 mIU/mL of follicle stimulating hormone (FSH) and 75 mIU/mL of luteinizing hormone at 6% CO_2_ and 37 °C for 24 h. If oocytes reached MII, oocytes were used for downstream experiments. Otherwise, immature oocytes with no polar body extrusion were kept in culture for a total of 48 h.

MediCult IVM^®^ System: LAG medium and IVM medium were pre-equilibrated at 6% CO_2_ and 37 °C for 12 h. Denudated immature oocytes were incubated in LAG medium for 3 h at 6% CO_2_, 37 °C. During this incubation, the final IVM medium (IVM medium supplemented with 1 mL of the patient’s own serum, 100 mIU/mL of hCG and 75 mIU/mL of FSH) was prepared. Immature oocytes were transferred to a new plate with the final IVM medium and kept at 6% CO_2,_ 37 °C for 28–32 h, after which their maturation state was assessed.

### 2.5. Oocyte Vitrification and Warming

Oocyte vitrification was performed with the MediCult Vitrification Cooling Kit (CooperSurgical^®^, Målov, Denmark) following the manufacturer’s instructions. Briefly a maximum of 2–3 oocytes were exposed to the equilibration medium until complete re-expansion. Oocytes were then transferred into the vitrification medium until vitrified, this procedure was performed within 1 min. In a minimal volume of medium, oocytes were loaded into the vitrification carrier and embedded into liquid nitrogen and stored.

Oocyte warming was performed with the MediCult Vitrification Warming Kit (CooperSurgical^®^, Målov, Denmark) following the manufacturer’s instructions, briefly the oocytes were quickly placed in the warming medium and the subsequent passages for dilution and re-expansion. At the final step, oocytes were washed and transferred to the culture medium and incubated for 2 h to be prepared for ICSI.

### 2.6. Spermatozoa Thawing

Sperm straw selected from a single sperm donor were thawed at room temperature, followed by washing and centrifugation. The supernatant was discarded and the pellet containing sperm cells was subjected to swim-up at 37 °C for 30 min. A sperm count was performed to assess suitability prior to IVF/ICSI.

### 2.7. ICSI and Embryo Development Observation

After denudation, IVM and warming, mature oocytes were placed in dishes with HEPES medium drops covered with oil, and a single sperm was injected into each of them under a microscope. The embryos were placed in culture dishes containing drops of the pre-equilibrated medium for embryo culture covered in paraffin oil and then incubated at 37 °C, 6% CO_2_ and 5% O_2_ until the blastocyst stage. Embryo assessment was performed with the time-lapse recording system. Fertilization was assessed 16–20 h after ICSI and annotated as normal when two pronuclei and two polar bodies were observed. Subsequently, the time-lapse system recorded images of each embryo, and embryo assessment was performed by analyzing the images in an external computer. Embryo morphology and development were checked and annotated every day. At the end of the culture (between day 5 and day 6), embryo morphology was scored once for a differentiated pattern of trophectoderm and the inner cell mass was clearly distinguishable. A minimum score of 5:2:2 and 6:2:2 was considered for defining an embryo as a blastocyst. Morphology assessment was established following the European Society and Human Reproduction guidelines, as indicated in the Atlas of Human Embryology [17].

### 2.8. Statistical Analysis

Statistical analyses were performed using the SAS software (version 9.4, SAS Institute Inc., Cary, NC, USA). Forty-nine GVs were required in each of the two pre-selected media groups, assuming a 30% maturation rate previously observed in the laboratory in order to find a 30% difference between the groups with 80% power in a two-sided test [18]. The likelihood-ratio, chi-squared test or fisher exact test were used. All tests were two-sided with a significance level of 0.05.

## 3. Results

### 3.1. IVM Culture Medium Selection

A higher proportion of mature MII oocytes was obtained in the SAGE group (86.0%) compared with the MediCult group (42.0%) (*p* < 0.0001) after 24 h of rIVM culture. No additional mature oocytes were observed after 48 h of culture. Thus, the SAGE^TM^ In Vitro Maturation Kit was used for the fresh vs. vitrified comparison in 2.1.2.

### 3.2. Fresh vs. Vitrified Comparison

A total of 43 (57.3%) mature oocytes were obtained from 75 GVs subjected to fresh rIVM and 13 (48.1%) MII oocytes were obtained from 27 vitrified and warmed GVs undergoing rIVM (Table 1, Figure 1). Thus, the likelihood of obtaining MII oocytes suitable for ICSI was slightly higher for fresh rIVM compared to immediate vitrification and subsequent rIVM after warming. The difference, however, was insignificant.

Fresh rIVM and ICSI yielded 11 (64.7%) zygotes properly fertilized and 5 (45.4%) of them developed until the blastocyst stage (Table 1, Figure 1). A total of 16 (61.5%) mature oocytes survived warming after vitrification and underwent ICSI, providing 4 (25.0%) zygotes from among which 1 (25.0%) reached the blastocyst stage. In the group of vitrified and warmed GVs subjected to rIVM 2 (20.0%) of 10 inseminated mature oocytes were fertilized but none of them progressed to blastocysts. Fertilization rates were significantly reduced in both the fresh rIVM-vitrification-warming-ICSI group (*p* = 0.0221) and the vitrification-warming-rIVM-ICSI group (*p* = 0.0248) compared to the control. A lower developmental potential for the blastocysts was observed in both groups utilizing cryopreservation; however, due to a low number of samples, no significant differences were observed.

## 4. Discussion

In general, our study showed that the surrounding conditions during immature oocytes IVM defined by the culture medium or manipulation steps of vitrification and warming can have a substantial impact on oocyte maturity, fertilization and developmental competence. In this study, we reported differences in the maturity rates of GVs collected from COS cycles when rIVM was performed using two different commercial protocols. A higher ratio of MII oocytes per GV oocyte was obtained using the SAGE^TM^ Kit compared with the MediCult IVM^®^ System. Thereafter, having selected SAGE medium for the exploration of new strategies for the fertility preservation of immature oocytes, we vitrified either GVs or the resulting MII oocytes, both before or after rIVM, respectively. Subsequently, every IVM-matured oocyte obtained was inseminated through ICSI and cultured until the blastocyst stage in order to evaluate their developmental competence. We observed that vitrification and warming carried out before or after rIVM compromised the fertilization and blastocyst development, compared with fresh oocytes undergoing rIVM and fresh insemination through ICSI. Previous studies reported an increase of 5.6% in live birth rates when rIVM is paired with IVF treatments [19]; furthermore, a live birth was reported after the IVF/ICSI of cryopreserved and warmed GVs which subsequently matured in vitro [16].

The clinical practice of using immature GVs for IVM from COS cycles has not shown clear benefits so far and there are doubts about its safety [11]. According to our results, rIVM coupled with vitrification has a negative impact on the meiotic states of both GVs (prophase I) and obtained MII oocytes. It shows a decrease in the fertilization rates and consequently seems to affect developmental progression as well. The slightly better outcomes observed in vitrified MII oocytes could indicate that vitrification is more deleterious in earlier immature states. Two previous works using GVs that were vitrified before and after IVM did not show an evident impact in the meiotic spindle constitution [20,21]. In particular, one of these works also reported a slightly detrimental effect on the ultrastructural organization of cytoplasmic components of mature oocytes generated from vitrified and warmed GVs [22]. In mouse models, it has been shown that nuclear maturation is commonly reached in naked oocytes under cycles primed with gonadotropins and that it does not seem to be affected by freezing–warming processes. However, the developmental potential is highly impacted compared to fresh GVs [23]. In light of these and our own results, it could be argued that oocytes acquire an endowment of cytoplasmic components needed for further embryo development and that these seem to be negatively affected by both denudation on early events and vitrification. So far it has been observed that during vitrification, high concentration of cryoprotectants induces subsequent osmotic changes of water and soluble components, which could drastically impact the cytoplasmic content. The cryopreservation of COCs is challenging because of the differential properties of oocytes and granulosa cells. Cryoprotectants cause the shrinkage of cells, which affects the gap junctions responsible for the bidirectional dialogue between both cell types and which are required for further progression to adequate maturation. To date, there is no cryopreservation method that is clinically suitable for maintaining an intact COCs structure, including both oocytes and cumulus cells, for further reproductive use [9].

In some patients, the use of immature oocytes may be a suitable option for maximizing the chances of fertility preservation and an alternative option to standard IVF. In PCOS women requiring IVF treatments, IVM may potentially reduce complications associated with ovarian hyperstimulation syndrome. A systematic review and meta-analysis indicated that the outcomes of using IVM in PCOS and non-PCOS patients showed comparable final outcomes [24]. IVM is also a proposed alternative for women with cancer who cannot undergo COS treatment for fertility preservation due to contraindication to undergo hormonal treatment or due to time constraints requiring the urgent initiation of chemotherapy [25,26]. In this context, performing IVM before vitrification showed a slightly higher likelihood of obtaining mature oocytes compared to IVM after vitrification. In addition, IVM before vitrification achieved a higher fertilization rate and better developmental competency compared to IVM after vitrification. Our study provides indications that may help to optimize the current strategies carried out for fertility preservation based on IVM. Despite the poor developmental outcomes of vitrified oocytes after rIVM, it could be inferred that non-stimulated cycles requiring IVM could be benefited when vitrification is performed in mature oocytes.

While the small sample size used in this study limits the strength of our conclusions, previous studies with similar results support our findings; in addition, since oocyte maturation and survival rates after warming may be patient-dependent, randomization performed by oocyte could have balanced this potential bias. Thus, further studies with a larger sample size are needed, as well as studies aiming at exploring the impact of cryopreservation at a molecular level, both on the cytoplasm and the genome at different maturity states. In future studies, we may also expand the applicable situations of this method, such as in patients who do not wish to undergo hormonal stimulation, which may also have an impact on the outcome of IVM culture.

In the current state-of-the-art of IVM, success is compromised by the patient’s condition and there are many uncertainties regarding the safety and suitability of this procedure. In particular, rIVM is associated with changes in the gene expression profile of human blastocysts [27,28]. Despite this, studies on epigenetic defects identified in children born after IVM are lacking, and this is thus a main concern in the clinical application of IVM [11]. The cytoplasmic components carried by fully mature MII oocytes are critical for processes released by fertilization, such as the maternal-to-zygote transition, the zygote genome activation and cell lineage establishment [29]. The cryopreservation of in vitro matured oocytes imposes a conditioning impact on the interacting elements in the cytoplasm that govern these biological processes and the further development of the embryo. Both cryopreservation and IVM are potential sources of changes in the epigenetic status of embryos, which could have consequences on the health of these future children.

## Figures and Tables

**Figure 1 jcm-11-01703-f001:**
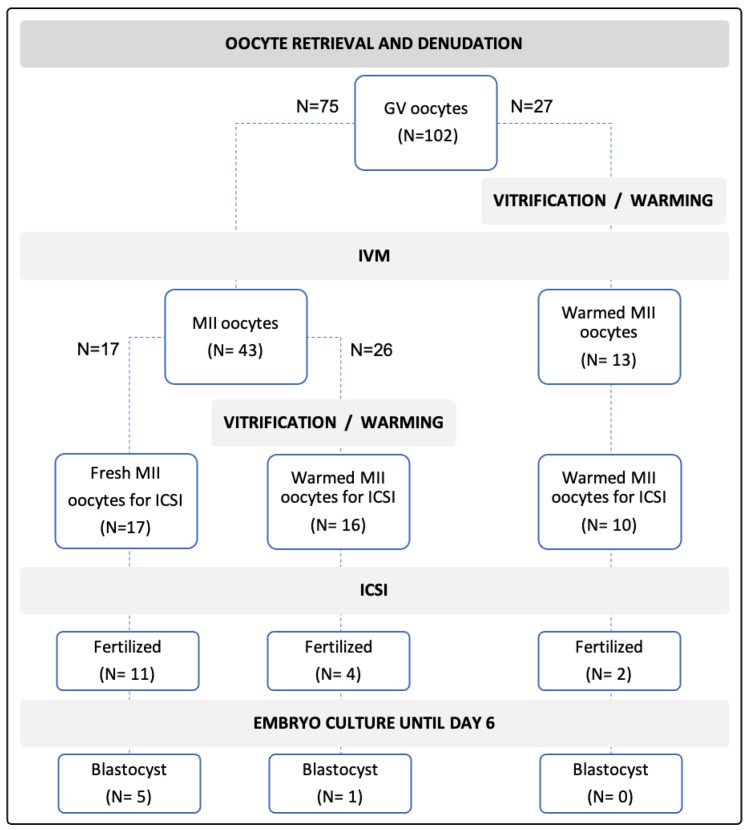
Overview of oocyte distribution along different processes of rIVM, vitrification and warming, and ICSI and embryo development.

**Table 1 jcm-11-01703-t001:** Outcomes of rIVM, vitrification and warming (V/W), and fertilization and blastocyst development after ICSI.

Outcomes	Control: Fresh rIVM-Fresh ICSI	Fresh rIVM- V/W-ICSI	V/W- rIVM-ICSI
Maturity rates	43/75 (57.3%)		13/27 (48.1%)
Survival after V/W	-	16/26 (61.5%)	NA
Fertilized/injection	11/17 (64.7%)	4/16 (25.0%) *	2/10 (20.0%) *
Developed blastocysts/fertilized oocytes	5/11 (45.4%)	1/4 (25.0%)	0%

NA: not available; *: comparing to Control, *p* < 0.05.

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
