# Peer review of "Exploring the Developmental Potential of Human Germinal Vesicle Oocytes Aiming at Fertility Preservation: Can We Increase the Yields of Competent Oocytes through IVM Combined with Vitrification?"

_jcm, 2022, doi:10.3390/jcm11061703_

Round 1

Reviewer 1 Report

Hao et al. provide a study on the developmental potential of rescue in vitro matured GV in combination with vitrification. The study is interesting as it analysis the competency of these rIVM oocyte up to the blastocyst stage and the latter information is largely missing in literature. 

remarks: 

1) title: 'the fertility potential': I would suggest using the word 'developmental potential'.

2) overall remark: oocytes are warmed after vitrification, not thawed. Please change this is the whole of the manuscript. Use: vitrified-warmed instead of vitrified-thawed. 

3) Material and methods: patient information is lacking. Although the study constitutes of 100 GVs (selecting IVM medium) and 102 GVs (vitrification hypothesis), it is important to know from how many patients these GVs were obtained from. A minimum of patient demographics can be added. 

4) 2.4 rIVFM of GVs: the rational here to use 2 IVM media is lacking. As I understood it well, the first experiment showed that the SAGE medium was superior, why are then 2 media used in this second step? 

5) Discussion: the blastocyst rate of the fresh rIVM-fresh ICSI is quite good.

It would be good to verify this % of blastocysts towards the % of blastocysts of the other oocytes of the patients. This would be very interesting to describe. 

Can the authors describe the quality of these blastocysts? Likewise, if timelapse was used, are there any developmental characteristics worth mentioning of analysing? When do the embryos then arrest, at which stage. It would be interesting to describe these observations. This information is lacking at the moment and could be of interest to the readers. 

Author Response

1) title: 'the fertility potential': I would suggest using the word 'developmental potential'.

Reply: Thank you for pointing out this, we agree with the reviewer and has changed ‘fertility potential’ into ‘developmental potential’ accordingly.

2) overall remark: oocytes are warmed after vitrification, not thawed. Please change this is the whole of the manuscript. Use: vitrified-warmed instead of vitrified-thawed.

Reply: We agree with the reviewer, the word ‘thaw’ has been replaced by ‘warm’ throughout the whole manuscript regarding oocytes.

3) Material and methods: patient information is lacking. Although the study constitutes of 100 GVs (selecting IVM medium) and 102 GVs (vitrification hypothesis), it is important to know from how many patients these GVs were obtained from. A minimum of patient demographics can be added.

Reply: Thank you for your comment. Indeed these data could be illustrative and we also realized the rationality behind recording that information. However, our ethical approval required the anonymization of the materials received, thus it is not possible to track on clinical characteristics. The general characteristics of patients treated at the center that provided the material for this research were briefly described in Lines 130-132.

4) 2.4 rIVFM of GVs: the rational here to use 2 IVM media is lacking. As I understood it well, the first experiment showed that the SAGE medium was superior, why are then 2 media used in this second step?

Reply: Yes, the first experiment was aimed at selecting among two IVM media, because both of them were available at our center at that time and we wished to select a further medium for IVM at our center as well. As it was described, ‘SAGE’ medium showed better results and was used in the second experiment. This has been clarified in the Material and Methods 2.1.1. where both media were compared and in 2.1.2, where SAGE was used alone. We hope our texts are more clear now. The reference to the brand ‘Medicult’ in 2.5, was for vitrification and warming, it was not used for IVM.

5) Discussion: the blastocyst rate of the fresh rIVM-fresh ICSI is quite good.

It would be good to verify this % of blastocysts towards the % of blastocysts of the other oocytes of the patients. This would be very interesting to describe.

Reply: Yes, this is indeed an interesting point to track. However, as previously explained, the clinical data of patients that gave their immature oocytes for this research did require the anonymization of the patients, thus is not possible to link our experiments to clinical information.

Can the authors describe the quality of these blastocysts? Likewise, if timelapse was used, are there any developmental characteristics worth mentioning of analysing? When do the embryos then arrest, at which stage. It would be interesting to describe these observations. This information is lacking at the moment and could be of interest to the readers.

Reply: We agree with the reviewer that it is an interesting point to go deeper into and it could provide extra information/parameters in evaluating the efficiency of different protocols. We have further clarified the scoring criteria applied in this study in Lines 190-195: In our study, blastocyst morphology was scored between day 5 and day 6 once a differentiated pattern of trophectoderm and inner cell mass was clearly distinguishable. A minimum score of 5:2:2 and 6:2:2 was considered for defining an embryo had become blastocyst. Morphology assessment was established following ESHRE guidelines as indicated in the atlas of human embryology.

The culture was limited until day 6 according to the ethical application. Additionally, due to the limited sample size, no further assessment was performed regarding the arrest moment.

Reviewer 2 Report

In the present study the authors conducted a prospective randomized study to investigate the fertilization and developmental potential of oocytes at the GV stage vitrified before or after in vitro maturation (IVM). The study is well designed and of high significance for patients undergoing fertility preservation strategies. The manuscript is clear and well written.

However, some minor revisions are suggested to the authors:

  1. The results are interesting but affected by a very small sample size. Authors should clarify how sample size calculation was performed to reach statistical significance
  2. The application of IVM to fertility preservation in cancer or PCOS patients should be discussed
  3. The possible epigenetic impact of performing IVM on GV oocytes should be discussed

Author Response

1.The results are interesting but affected by a very small sample size. Authors should clarify how sample size calculation was performed to reach statistical significance.

Reply: The power calculation has been described in detail in the Statistical Analyses in Lines 201-204.It is also mentioned in the Material and Methods section 2.1.1.

2.The application of IVM to fertility preservation in cancer or PCOS patients should be discussed

Reply: Thank you for your comment. We have now included a paragraph referring to the application of IVM in PCOS and cancer patients who will benefit from IVM in the introduction (Lines 57-61) and the discussion (Lines 284-294).

3.The possible epigenetic impact of performing IVM on GV oocytes should be discussed

Reply: We have included a paragraph at the end of the discussion section indicating the still unknown epigenetic defects that could be associated to IVM of oocytes, as suggested by the reviewer, in Lines 314-317.

Round 2

Reviewer 1 Report

The authors, have addressed the comments that were raised.

It is a pity that their ethical board obliged the researchers to work with completely anonymized samples and not pseudonomized. The work could have contained more interesting data.